# Total suspended matter derived from MERIS data as indicator for coastal processes in the Baltic Sea

D. Kyryliuk<sup>1</sup>, S. Kratzer<sup>1</sup>

<sup>1</sup>Department of Ecology, Environment and Plant Sciences, Stockholm University, Stockholm, 106 91, Sweden

5 Correspondence to: D. Kyryliuk (dmytro.kyryliuk@su.se)

**Abstract.** Total suspended matter (TSM) is an indicator of coastal processes and can be retrieved reliably from MERIS (Medium Resolution Imaging Spectrometer) data. In this project we used MERIS TSM data from a Swedish coastal monitoring system <u>www.vattenkvalitet.se</u> to evaluate the physical extend of coastal processes.

The data set consisted of all viable MERIS scenes during summer (June-August) 2009-2011, covering the whole Baltic Sea area. Monthly composite images were produced for each year, and the monthly composites were subsequently evaluated with regards to terrestrial influence, and the typical features caused by cyanobacteria blooms (typically during July and August).

Next, a composite image from early June 2011 was generated in order to exclude a possible influence from cyanobacteria on the patterns of TSM distribution. This early June composite was then used as a basis to evaluate the extent of terrestrial

influence using the NW Baltic Proper (Swedish coastal areas) and the SE Baltic Proper (Latvian, Lithuanian and Polish coastal waters) as examples.

In both areas the averaged TSM scenes from early June were used to extract transects of TSM data perpendicular to the coast, spanning from coastal to offshore waters. Based on previous bio-optical research in the NW Baltic Sea and on further statistical analysis of MERIS scenes local coastal water thresholds were defined for different areas in the Baltic Sea. Using

- these local thresholds, it was found that coastal processes in the NW Baltic Sea extend to approximately 15-27 km off-shore, whereas in the SE Baltic Sea the coastal influence extended to about twice the distance i.e. to about 34-52 km off-shore. Next, the trendlines of TSM distribution along transects was evaluated mathematically. The trend line for the NW Baltic proper was best described by a polynomial equation, whereas the trend line from the transect in SE coastal waters was best described logarithmically in areas of high resuspension. These differing trends indicate different hydrological regimes in the
- two areas, which are mostly driven by a combination of land run-off distributed by diffusional processes, and coastal dynamics driven by local wind exposure. The results demonstrate that ocean colour remote sensing can provide important information for Baltic Sea research and management, as well as for the monitoring of coastal processes. The method allows for an evaluation of the extent of coastal influence, and of seasonal fluctuations in river run-off and phytoplankton dynamics. Furthermore, the concentrations of total
- suspended matter in the different sub-basins of the entire Baltic Sea can be compared synoptically.

#### **1** Introduction

Coastal waters are highly influenced by coastal run-off and by nutrients from land, stimulating algal and phytoplankton growth (Lessin and Raudsepp 2007). The coastal zone also provides habitats for mariculture and nurseries for fish (Kratzer and Tett, 2009), and has a high recreational value for humans. It is also the area that is most strongly affected by pollutants

from land (Cederwall et al. 1990).

For monitoring the water quality status and for management it is important to define the extent of coastal areas and processes. The definition of the coastal zone, however, is a complicated task in the Baltic Sea as there is no clear topographic feature, such as the continental shelve, delineating different water types. The most common definitions for coastal waters are usually based on physical criteria. For example, the coastal zone may be defined as the area where the water depth is less

- than a fixed value. This approach makes e.g. near-shore and island water bodies part of the coastal zone (Leppäranta, 2009). For management purposes the EU Water Framework Directive (ECE, 2000) defines the coastal zone as: "the surface water on the landward side of a line every point of which is at distance of one nautical mile (nm) on the seaward side from the nearest point of the baseline from which the breadth of territorial waters is measured". This means that it is defined here as a fixed distance to the shore. In reality, however, the coastal zone is larger and may reach much beyond 1 nautical mile as can
- be seen in satellite images.

The Swedish national definition of the extent of the coastal zone is mostly based on physical criteria and a hydrological model developed by the Swedish Meteorological and Hydrological Institute (SMHI). The physical parameters used as input into SMHI's model are the maximum depth, salinity, stratification, wave exposure, number of days with ice cover, as well as bottom substrate. The model is used for classifying all Swedish coastal waters, and for dividing them into different water bodies that are used by HELCOM for the periodic ecological status assessment (Naturvårdsverket, 2006.).

- Coastal areas are rather dynamic systems and require a monitoring approach that can capture the dynamic processes in coastal waters. For example, storm events can lead to a sudden sediment bloom, and a phytoplankton bloom can develop within several days or weeks. These dynamics can be monitored more efficiently by using data derived from ocean colour satellites (Attila et al., 2013; Kahru et al., 2014; Kratzer et al. 2014; Harvey et al., 2015).
- The Medium Resolution Imaging Spectrometer developed by the European Space Agency (ESA) and launched on ENVISAT in 2002 was the first operational ocean colour sensor that was specifically adapted to monitor coastal zones (Schiller and Doerffer, 1999). Because of its improved spatial and spectral resolution, it was capable to sense the dynamical changes in coastal water quality, and it also has been shown to improve the temporal resolution of water monitoring programs (Kratzer et al, 2014; Harvey et al., 2015), even in areas with relatively frequent cloud cover, such as the Baltic Sea

(Isemer and Rozwadowska, 1999; Karlsson, 2003). One parameter that can be used to identify the extent of the coastal zone is total suspended matter (TSM), which also can be detected from space using ocean colour remote sensing data. Previous bio-optical research has shown that TSM may be used to evaluate the extent of the coastal zone in in-water Baltic Sea waters that are optically dominated by CDOM. Kratzer and

Tett (2009) developed an optical model that was used to verify a dynamic model describing the distribution of matter in a coastal area. This model assumed diffusion to be the driving force for the distribution of suspended matter, given that tidal influence in the Baltic Sea is almost absent. It was found that diffusional transport of bio-optical variables along an axis perpendicular to the coast can be described using a steady-state model. According to this model the distribution along the

- axis can be characterized as a low-order polynomial (order 1-3) when particles are transported from an inland source (e.g. sewage treatment plant) into the open sea. The derived optical model defined the extent of the coastal zone in the range of 15 km for north-western Baltic Sea during the summer period, based on inorganic suspended matter. The authors therefore hypothesized, based on bio-optical in-water measurements that the extend of the coastal zone in the NW Baltic proper should be in the range of tens of kilometres, taking seasonal variations into account, and not 1 nm as defined by the WFD.
- TSM can be retrieved with relatively high accuracy from MERIS data (Kratzer and Vinterhav, 2010; Beltran et al., 2014), i.e. with a bias of about 8-16%, dependant on the processor used. In this paper we transpose the idea of the one- dimensional bio-optical model described in Kratzer and Tett (2009) into a two-dimensional model using MERIS data, covering the whole Baltic proper.

The main objectives of this study are to 1) to generally derive mean estimates of total suspended matter load of the different

Baltic Sea sub basins using MERIS data, 2) to investigate if MERIS data can be used to evaluate the extent of coastal processes in the Baltic Sea and 3) to compare the extent of the coastal water masses as derived from remote sensing data to the water body classification defined by SMHI, with Baltic Sea bathymetry and marine seabed sediment.

## 2 Methods

Full Resolution (FR) level 2 MERIS data was provided by Brockmann Geomatics AB. The data had been processed by
Brockman Consult (BC), Germany for the CoastColour project and was delivered to the Swedish coastal monitoring system (vattenkvalitet.se). The delivered level 2 data had been processed the following way. MERIS full-resolution level 1b data (v.3) was geo-corrected using AMORGOS. This was further processed to level 1p, including radiometric and smile correction, as well as a correction for adjacency effects using ICOL (Santer et al., 2009). A pixel identification tool developed by BC was used to mask clouds and mixed pixels. Level 2 data was subsequently generated from level 1P using
the Case-2 Water Properties processor from Freie Univesität Berlin (FUB processor, Schreoder et al., 2007).

The provided level 2 data covered the entire Baltic Sea region and consisted of all viable MERIS scenes during the summer periods (June-August) 2009-2011. These had already been identified by Brockmann Geomatic AB for inclusion in the Swedish coastal monitoring system. **Figure** 1 shows a summary of all scenes that were used for the binning procedure (BEAM software) to derive monthly composite images.

## 2.1 Areas of investigation

During pre-analysis two contrasting areas were chosen each for a case study to evaluate the extent of the coastal zone: the NW Baltic and the SE Baltic proper (**Figure 2**). The areas were chosen based on differences in run-off, depth, wind exposure and bottom substrates (Kratzer et al., 2011).

#### 5 2.2 Geolocation

The data used in this project has been geolocated with AMORGOS (Accurate MERIS Ortho Rectified Geo-location Operational Software), developed by ACRI-ST in France. AMORGOS includes a precise orbit determination, instrument pointing and performs an ortho-rectification. Evaluation of AMORGOS processed data compared to manually geo-corrected data was done prior to the study using Ground Control Points in ortho photo images available in digital format from the

10 Swedish National Land Survey. For most evaluation points, AMORGOS generated slightly better accuracy, and a much better overall quality of the correction was achieved.

#### 2.3 Radiometric correction

Data needs to be radiometrically calibrated, which initially means that raw data is converted from digital numbers to Top-Ofthe-Atmosphere (TOA) calibrated radiances measured in mWm<sup>-2</sup>sr<sup>-1</sup>-nm<sup>-1</sup>. In addition, throughout the lifetime of a sensor the 15 detectors age, which needs to be accounted for by applying a radiometric correction model and coefficients.

#### 2.4 L2 processing

L2 processing consists of atmospheric correction and retrieval of concentration of the water constituents. During recent years, a number of coastal and inland L2 processors have been developed besides the standard MERIS/ESA processor (MEGS) and results have been improved. Further development is an ongoing research topic. Known examples are the Case 2

- 20 Regional Processor (C2R) Doerffer and Schiller. (1999, 2007), the Boreal Lakes Processor (BOR) (Doerffer and Schiller, 2008) and the Eutrophic Lakes Processor (EUL) Koponen et al. (2008) and Ruiz-Verdu et al. (2008), all three processors developed by HZG, Germany and using identical atmospheric correction algorithms. However, they all have different water quality algorithms, each adapted to a certain water type. Along with these aforementioned processors, the FUB coastal Water Properties Processor from Free University Berlin Schroeder et al. (2007) has also been tested for the Baltic Sea
- 25 (Kratzer and Vinterhav, 2010; Beltran et al., 2014), the Gulf of Bothnia and for Lakes Vänern, Vättern and Mälaren. FUB generated the best results for all areas and has therefore been used for implementation in the coastal observational system www.vattenkvalitet.se.

## 2.5 L2 Mosaic function

The entire Baltic Sea is usually not covered by one single MERIS overpass. In order to derive a full image of the Baltic Sea one has to make a mosaic composite image of at least 2 MERIS scenes. Figure 3 shows how the mosaic function in BEAM that is used to generate a mosaic composite image.

## 5 2.6 L3 processing

In order to evaluate the extent of the coastal processes and to see seasonal variability in TSM concentration in the Baltic Sea L2 mosaic images for each month were used. The L3 binning function was used to derive averaged monthly mean images (composites) and the spatially and temporally binned products are referred to as L3 data. 'Binning' refers to the process of attributing the contribution of all level 2 pixels in satellite coordinates to a fixed level 3 grid using a geographic reference

system (Figure 4). A sinusoidal projection is used to realize a Level 3 grid comprising a fixed number of equal area bins with global coverage.

Monthly means (level 3 products) of the TSM concentration were generated and subsequently used to evaluate the extent of the coastal zones along the whole Swedish coast. The aim was to evaluate if there was a difference in the extent of the coastal zone for different areas and different seasons of the year, which may be caused by differences in precipitation or wind emograph.

wind exposure.

## 2.7 Using TSM to evaluate the extent of coastal processes

*In situ* TSM is usually measured gravimetrically and can be divided into an organic and an inorganic fraction (Strickland and Parson, 1972). Kratzer and Tett (2009) showed that inorganic TSM can be used to identify the break from coastal to open sea water, and the authors used a coastal threshold of 0.05 gm<sup>-3</sup> for inorganic TSM, which corresponded to about 15 km off-

- shore. However, inorganic suspended matter is not a standard level-2 product for MERIS. The trend lines in Kratzer and Tett (2009) show that the threshold for TSM should be in the range of about 0.8 (+/- 0.3) g m<sup>-3</sup> in the NW Baltic Sea, corresponding to the distance from the shore where the concentration of inorganic matter also tends towards zero. Using individual MERIS scenes several coastal areas in the Baltic Sea were evaluated with regards to a general coastal threshold for the Baltic Sea. A rough, preliminary statistical analysis showed that 0.6 g m<sup>-3</sup> is an overall more appropriate threshold for
- different Baltic Sea basins, which is also within the mean and standard deviation of the local threshold identified for the NW Baltic Proper. A similar threshold of 0.5 g m<sup>-3</sup> was defined by the NASA SeaWiFS protocol (Mueller and Austin, 1995), to generally identify the transition from Case-1 to Case-2 water masses. As Case-1 a water mass is categorised if total SPM (TSM) concentration is less than 0.5 g m<sup>-3</sup> (dry weight).

Next, the TSM concentration extracted from MERIS data was used to evaluate the trend of TSM along a gradient from 30 coastal to offshore waters in the two Baltic Sea areas chosen for this study (Figure 2). It was evaluated if transects extracted

perpendicular to the coast follow a polynomial decline and if a clear break could be observed between coastal and open sea waters, and at what distance to the shore.

## 2.8 TSM concentrations for different sub basins

The preliminary analysis using MERIS images showed that the concentration of total suspended matter varies somewhat in 5 the different sub-basin. In order to evaluate the concentrations for each sub basin particularly in the region of interest (area A and area B), the HELCOM classification and division of sub basins in the Baltic Sea was applied. This included both coastal and offshore water bodies. The interactive map is available at <u>HECLOM map service</u> and includes shapefiles and background information about how the sub-basins were defined (HELCOM Monitoring and Assessment Strategy, 2013). The HELCOM sub-basin definition was further applied to the early June composite from 2011, where TSM values on the

10 two regions of interest were extracted using vector shape files (Figure 5 and Figure 6).

## **3** Results

The resulting binned time-series of TSM concentrations (monthly means) during the summers 2009, 2010, 2011 are shown in (Figure 7).

#### 3.1 Excluding the influence of cyanobacteria occurrence

- The binned products (Figure 7) were evaluated with regards to terrestrial influence and the typical features caused by cyanobacteria blooms occurring typically during July-August (Hajdu et al., 1997; Kahru et al., 2007). The monthly composite from June 2011 (image 2011-06) in Figure 7 seemed least influenced by cyanobacteria blooms. However, Kahru and Elmgren (2014) found recently that the timing of surface accumulation of cyanobacteria in the Baltic Sea has shifted earlier in the season, by approximately an average rate of 0.6 days per year, causing the peak of the summer blooms to occur
- about 20 days earlier when compared to the start of the satellite measurements in 1979. In order to avoid the influence of cyanobacteria blooms in the TSM product, it was therefore decided to bin only images from the first part of June (Figure 8) and to use this composite for further analysis. This early June composite still shows the typical currents that are usually indicated by blooms of filamentous cyanobacteria. However, at this time of year, cyanobacteria blooms are unlikely to occur, and therefore it can be assumed that the image actually shows the distribution of coastal inorganic sediments, and therefore 25 can be used to assess the extent of coastal processes in the Baltic Sea basin.
- In order to evaluate the distribution of phytoplankton in early June chlorophyll-a composite images for each year (2009-2011) were derived and compared to the TSM composites from early June (Figure 9). This comparison showed that the chl-a distribution in early June illustrates a different pattern when comparing to the TSM distribution, indicating a certain degree of de-coupling of the two parameters. It can therefore be assumed that the early June TSM composite does not represent the
- distribution of filamentous cyanobacteria, but instead indicates the distribution of inorganic sediments, and thus can be used

to evaluate the extent of coastal processes. The chl-a images show a phytoplankton bloom in the Bothnian Sea in 2010, which is most likely a late phytoplankton spring bloom. In the Baltic proper the chl-a concentrations are rather low (presumably indicating the occurrence of small flagellates), which is to be expected after the spring and before the onset of the summer bloom of filamentous cyanobacteria.

## 5 3.2 Evaluating the extent of coastal processes in comparison to standard methods

In order to evaluate how the SMHI-defined water body classification compares to the TSM concentration along the coast, the SMHI shape file 'Havsomr\_y\_2012\_2' was overlaid on top of TSM composite image from early June 2011 (Figure 10 a), given that this composite had the best coverage for the Baltic Sea and was also composed of 3 images. Each pixel contained a value representing the average concentration of total suspended matter in g m<sup>-3</sup>. The MERIS TSM composite from early

- June 2011 overlaid with the shape file defining coastal zones according to SMHI shows that the coastal influence in reality reaches much further offshore in the eastern and southern areas of the Baltic Sea than both indicated by the SMHI water classification, and the 1 nautical mile line as defined by the EU WFD. The SMHI definition looks here rather arbitrary and fixed compared to the TSM distribution, but provides reference borders since the classification is based on physical criteria. The north-western and south-eastern parts of the Baltic Sea (Figure 10 b and 10c) were further analysed. Trend analysis was
- performed in order to evaluate the behaviour of particles along a transect from coastal to offshore waters. During the analysis of the trendline it was found that most of transects were best described either by a logarithmic or a polynomial function. Thus, both functions were plotted on each figure, choosing the order of polynomial that scored the highest r-squared (Figure 11a).

The same analysis was applied to Bråviken; a bay situated about 50 km south-west of Himmerfjärden bay. Bråviken shows

- much higher TSM concentrations and has a higher gradient of TSM, and the threshold was reached here at about 15 km offshore (about 50 km from Norrköping) for the polynomial function that showed the highest r-square value (Figure 11b). Along the eastern, south-eastern and southern coasts, the trendlines were described best by either using a logarithmic function, or polynomial regression. Here, the threshold was reached much further off-shore, at about 34-52 km from the coast (Figures 11a Figure 11b)
- The values for TSM concentrations are slightly lower in the north-western basins of the Baltic proper (if not significantly influenced by cyanobacteria blooms), whereas in the eastern and southern basins concentrations are higher, presumably due to the higher input from large rivers in the southern Baltic (i.e. the Vistula and the Oder) and the sediment transport along the eastern coast up north due to the Coriolis force. The rivers bring in larger amounts of anthropogenic load from the densely populated areas in Poland and Germany. Higher concentrations of TSM in the water column may also be due to difference in
- bottom type (less exposed rock than at the NW Swedish coast), and due to mud and sand kept in suspension by wind stirring and the dynamics of surface currents. Table 1. summarizes the results of the trend analysis.

The monthly mean concentrations of TSM (June 2011) in all sub-basins of the Baltic Sea were extracted from MERIS data. The histograms of the composite images show means of TSM concentrations for the different Baltic Sea sub-basins are shown in Figure 12. The ranges of values and statistics for the different basins are shown in Table 2.

## 4. Discussion

- 5 The delineation of coastal waters using the water body classification by SMHI has shown to have only little agreement with the MERIS L3 binned TSM product for the Baltic Sea. In the NW Baltic, the TSM concentration reaches the local threshold of about 0.57  $g m^{-3}$  between coastal and transitional waterbodies at a distance of approximately 15-27 km offshore. In the E, SE and S Baltic the situation is different. In the of those transects (E1, SE2, SE3, S4) we can observe that the TSM concentrations reaches the respective local thresholds ranging from 0.59 to 0.79 gm<sup>-3</sup> at a distance of approximately 34-52
- 10 km which is almost twice the distance compared to the north-western coast. The seabed in the southern Baltic is dominated by inorganic suspended matter (sand and clay), and the TSM particles extend much further into the open sea. In the NW Baltic Sea, the sea bottoms are dominated by exposed rock (Figure 13). It was rather interesting to observe that the trend distribution of TSM particles along defined transects, in the E, SE and S

Baltic Sea was not only best described by polynomial regression of the second order, as found by Kratzer and Tett (2009) for

- north-western Baltic, but could be also well described by a logarithmic function. The highest coefficient of determination was used to identify which function had a better fit. The trend for both NW1 and NW2 were best described by a polynomial regression of  $2^{nd}$  order with  $r^2 = 0.84$  and  $r^2 = 0.95$ , respectively, implying that here diffusion is the dominant driver behind the particle distribution (Kratzer and Tett, 2009). As for the E, SE and S Baltic Sea the trend analysis was more complex, since other physical forces take place along this part of the Baltic Sea coast. Here, it can be assumed that wind-driven
- resuspension combined with the Coriolis effect are the main processes for distribution of TSM particles away from the coast, all the way into the open sea. In prior transect extraction from the composite of early June 2011, it was identified by Kyryliuk (2014) that the distribution of TSM particles along the transect in the south-eastern Baltic Sea using monthly composite from June 2011 could be described well by a logarithmic function, and overall had a better fit. However, our recent results from early June 2011 have shown that for the transects E1 and S4 the trend could be best described by a
- logarithmic function as assumed for the south-eastern coast. But transects SE2 and SE3 can also be described by a polynomial of second order, which is more typical for the north-western Baltic Sea. These differences could be caused by the high variability in coastal processes. For instance, a logarithmic trend may imply changes in the coastal dynamics (E1 and S4), whereas for SE2 and SE3 off-shore resuspension caused by eddies (fluid dynamics or two currents creating mesoscale features) may change the trendline from logarithmic to polynomial.
- Essentially, the bio-optical model for north-western Baltic Sea (Kratzer and Tett, 2009) is based on a steady-state model that assumes a diffusional transport along the axis perpendicular of the coast. The mathematical solution comes from the use of a series of power functions that led to a comparison of simple polynomial models of various cases (1<sup>st</sup>, 2<sup>nd</sup>, 3<sup>rd</sup>), where the

transport of particles is moving from the source (land, river outlet, sewage treatment plant; high end member) towards the open sea (sink or low end member). Here, low order polynomials denote various biological and chemical processes within a given gradient. Kratzer and Tett (2009) found that SPM (conservative substance) was best described by a second order polynomial, CDOM by third order polynomial and chl-a by a linear function. As for the south-eastern Baltic, it is possible to

- derive a steady-state exponential regression model based on the assumption that (1) there is a coastal source and an off-shore sink; (2) that the transport distance of SPM that moved offshore is a linear function of time elapsed since a packet of water left the source; (3) that the regression rate is constant. However, at the south-eastern coast of the Baltic Sea a logarithmic function showed a better fit, indicating different dynamical processes. Another explanation why TSM is carried further offshore in this region may be found in the bottom topography of the Baltic Sea. The SE Baltic is rather shallow and slopes
- rather smoothly, which has an effect on the extent of physical processes such as coastal upwelling (Myberg and Andrejev, 2003) and the development of internal waves. The SE Baltic Sea is also more exposed to wind (Danielsson et al., 2007), which combined with the sandy and muddy bottom substrates, enhances the dynamics and the resuspension of sediments, and the transport of particles further offshore.

The NW Baltic is less exposed to wind and therefore diffusion may be assumed here to be the main driving force for the

- distribution of matter as assumed and tested in Kratzer and Tett (2009). Furthermore, in the NW Baltic proper, the bottom depth increases within a relatively short distances from the shore. i.e. coastal morphology is much steeper (Figure 15). The deepest part of the Baltic Sea is at Landsort Deep (459 m depth), which is only about 30 km offshore. The bottom topography is also characterised by a great number of sills, separating different sub-basins (Figure 15 NW1). These factors combined may lead to the settling and falling out of suspended matter closer to the shore, i.e. the sills interrupting the
- different basins may prevent exchange with the open sea water, and the internal circulation in each basin may facilitate the settling out of heavier particles closer to the shore.

The water depth (m) values of the transect NW1 through Himmerfjärden bay and out to Landsort deep were plotted against the distance from the shore (in km). This depth profile was then compared to the TSM values extracted along the same transect. The depth profile was obtained from The Baltic Sea bathymetry by HELCOM and IOW (<u>maps.helcom.fi</u>). The

- shoreline slopes are quite deep and erratic in the Himmerfärden area (Figure 15). Himmerfjärden consists of a series of basins that are separated by sills. The depth profile in Bråviken (NW2) shows a different trend. The water depth remains rather shallow when moving away from the inner bay which is also interrupted by sills. The increase in depth becomes obvious at 40-45 km, at the geographical 'end' of Bråviken. Here, the TSM concentrations drop rapidly and tend towards local threshold (0.57 g  $m^{-3}$ ) at approximately the same distance as the sudden increase in depth. The threshold is reached at
- a depth of about 15-20 meters. The eastern transect (E1) at the Latvian coast shows yet another depth profile. Here, the shoreline slopes quite smoothly when compared to NW1 and NW2, without any intermittent sills. The slope is rather steep in the first 10 km with a brief transition to lower depths and then the depth drops suddenly between 25 km (at 15 m depth) to 50 km (at 90 m depth). This increase can be described as gradual, but relatively steep. The observed TSM values follow the depth profile closely to up about 10 km distance from the shore, after which no clear pattern of co-dependence is detected.

The SE2 depth profile is steep but gradual, without any recurring intermittent sills. The TSM values along the transect nearly repeat the pattern of the depth profile starting to decline at about 10 km from the shore. For the SE3 profile the increase in depth is linear and rather steep, again, with no interrupting sills but with a few ridges. The TSM values drop here suddenly within the first 5 km where the depth also drops from 0 to 20 m. Further at about 10 km distance the TSM concentration gets

- higher and then drops again, tending towards the local threshold and following a polynomial trend as the depth increases again. The southern transect (S4, Gulf of Gdansk) the TSM concentration drops suddenly at about 10 km off the shore, where the depth also decreases to 60 m. Thus, both the SPM concentration and the depth profile show a similar trend. After 15 km the bottom inclination becomes more gradual. The TSM values show a logarithmic trend along the chosen transect. Physical forces such as diffusion, wind, resuspension and more importantly bottom profile and the nature of the bottom
- substrates may thus play a significant role in distribution of total suspended matter in coastal regions. Similar sediment dynamics were observed by Tang et al. (2013) on the Beaufort Shelf, which is influenced by the Mackenzie River run-off, wind direction and the surface circulation. The authors analysed the dispersion pattern of the Mackenzie River plume near the mouth of the estuary, using algorithms for two highly contrasting run-off scenarios. The first scenario included a similar wind forcing strength when river run-off was low. Here, the sediment plumes tended to spread less further offshore and with
- lower TSM concentrations within the plumes. The second scenario was linked with higher river run-off, where the entire coastal area was subsequently loaded with very high TSM concentrations, leading to an occurrence of extensive offshore plumes. Tang et al. (2013) concluded that remote sensing can indicate important aspects about the dispersal pattern of TSM plumes, and the distribution of suspended matter in coastal ecosystems.

Mulligan et al. (2010) observed that not only river run-off can lead to rather different plume distribution patterns, but also wind direction and physical forcing. A combination of locally-derived optical algorithms for deriving TSM from remote sensing data allows for a rather comprehensive overview of coastal dynamics in relation to environmental forcing. Increased precipitation leads to increased land run-off and fluxes of sediment loads to the sea (Miller and McKee, 2004), and TSM generally acts an indicator for coastal run-off and wind forcing (Kratzer and Tett, 2009).

An increase of land run-off has been predicted for Northern Scandinavia with the advance of climate change. Time series of remote sensing data can be used to evaluate changes in the distribution of suspended matter in coastal areas, which are most affected by river discharges in the vicinity of urban areas. The algorithms developed for regional applications play a significant role in acquiring reliable measurements from satellite images. Further research requires the use of more accurate regional algorithms for TSM retrieval as well as adapting hydrological models to be able to estimate water exchange with the coast and how climate and certain weather conditions e.g. wind direction and precipitation can affect concentration of suspended particles.

#### **5** Conclusions

This work demonstrates that ocean colour remote sensing can provide important information for Baltic Sea research and management e.g. from the investigation of river discharge and phytoplankton blooms to the monitoring of coastal processes. The main advantages of marine remote sensing methods are synoptic large-scale coverage of the entire Baltic Sea (Figure 6-11) with good spatial and temporal resolution (Kratzer et al., 2014; Harvey et al., 2015).

- From a management perspective it is rather crucial to understand how ecosystem state varies over time and space and to have a synoptic overview of these changes. MERIS images as shown in Figure 8 allow for the monitoring of diffuse as well as point sources, such as a sewage treatment plant. This is important, as usually only the big streams are monitored by the Swedish river and coastal monitoring programs (SLU database).
- TSM has shown to be a good indicator for ecosystem state, as it both indicates terrestrial run-off as well as wind-driven resuspension of particles, which is typical for coastal processes. Monthly mean estimates of TSM load for the different Baltic Sea basins can be used for evaluating seasonal fluctuations in those physical drivers. This is important complementary data for the monthly means of chlorophyll, which are used, both in the HELCOM periodic assessments as well as for the definition of water quality status according to the WFD. Turbidity (caused by SPM) is one of the mandatory parameters
- listed in the EU's Marine Strategy Framework Directive (Annex III, EC, 2008). Thus, including remote sensing in management allows for a more comprehensive analysis of system drivers and the productive status of the marine ecosystem. In this paper we used local thresholds between coastal and offshore waters ranging from 0.57-0.79 mg m<sup>-1</sup>, based on the statistical analysis of MERIS data and the HELCOM division of sub-basins. The results indicate that the coastal zone extends substantially farther than shown by the water body classification by SMHI.
- Here, satellite-derived TSM has shown to be an alternative tool for describing the fluctuations and changes in the extent of coastal processes. Strictly speaking, the extent of coastal waters is not a fixed line in relation to the distance from shore but it varies with seasonal variations of terrestrial run-off and local weather and/or climate conditions, such as sudden storm events as well as general high wind exposure as in the SW Baltic proper, where there is also a high fluvial input. More work has to be done to evaluate the seasonal fluctuations of the coastal zone, and also to evaluate the uncertainties in the method.
- TSM can be derived with rather high accuracy from MERIS data (Kratzer and Vinterhav, 2010; Beltran et al., 2014), and may therefore be advantageous in comparison to a model that is based on many input variables, such as the hydrological model developed by SMHI. Each variable has its own error attached to it, and all errors add up to the total error. It is also known from methods in ecology that the more a complicated method, the higher the uncertainties inherent in the method (Peters, 1991).
- Furthermore, the water-exchange model developed for SMHI was initially developed on hydro-geophysical parameters in Himmerfjärden (Engqvist 1996), and was then later applied to the whole Swedish coastline. However, the hydrological schemes and sediment transport into the different catchments may differ, as shown in this paper, causing regional errors.