# Peer review of "Total suspended matter derived from MERIS data as indicator for coastal processes in the Baltic Sea"

_Ocean Science, 2016_

## Referee Comment (RC1) · Anonymous Referee #1 · 23 Feb 2016

This manuscript is very descriptive in nature. It contains many figures and tables and the manuscript is mainly arguing why the MERIS images shown contain information on the "physical extent of coastal processes". However, I found very little substance in the manuscript that explains how this methodology works and how it can be tested or validated. A proper working hypothesis is lacking and when we look at the objectives of the study (page 3, Lines 14-17) none of them is properly reached;

"1) to generally derive mean estimates of total suspended matter load of the different 15 Baltic Sea sub basins using MERIS data" The methodology to process and combine MERIS data is standard. In figure 1 a list of 25 images is presented. Then follows a very obscure selection process that removes 22 images and only 3 images in June

were selected. How this was done and why all the others (these show very different TSM distributions after processing) were troubled by cyanobacteria blooms is not clear from this paper.

"2) to investigate if MERIS data can be used to evaluate the extent of coastal processes in the Baltic Sea."

Section 2.7 gives a description of the TSM concentration threshold that marks the transition between coastal waters and non-coastal waters. Besides a reference to an "inorganic threshold of 0.05 gm-3 (Kratzer and Tett, 2009) no explanation is given why this can be translated to TSM values of 0.6 gm-3. What does it mean? What are processes? Why TSM only? What are the arguments to use the TSM value at the peak-frequency of the TSM distribution (Figure 6) as transition value. Maybe the authors have good arguments, but they are not given and the reader has no way to test or interpret these results.

"3) to compare the extent of the coastal water masses as derived from remote sensing data to the water body classification defined by SMHI, with Baltic Sea bathymetry and marine seabed sediment." It is nice that in a previous study the results of MERIS pro-cessing was extensively described and the algorithms tested and validated (Beltran et al, 2014). From this study it is clear that the typical error in TSM retrieval is at least in the order of 30% and the bias (stated on page 3, Line 11) is not the dominant error. With this uncertainty in mind, the preference for a certain analytical function to describe the TSM transect values as function of distance from the coast becomes doubtful.

Finally, the authors have provided illustrations of the seabed sediment (Fig. 13) and bathymetry (Fig 14) in order to illustrate points in the discussion. However, this infor-mation is not new and was not used in the methodology and result section.

---

## Referee Comment (RC2) · D. Bowers (Referee) · 21 Mar 2016

Remote sensing of coastal seas from space has provided us with some interesting observations leading to changes in the way that we understand shelf seas to work. For many shelf seas, especially in northern Europe, the strongest signal we can see in satellite imagery is the distribution of near-surface suspended sediments. A strength of this paper is that it uses this strong signal to tackle an important management issue: can we identify how far out to sea the effect of the coast extends? The paper builds on earlier work which predicts that the suspended sediment concentration should decay with distance offshore and the offshore extent of 'coastal processes' can be defined as the line marking where the suspended sediment load falls below a critical value.

[Figure]

There are two weaknesses with the paper, as I see it, which should be addressed by the authors. The first is that, I am not sure that suspended sediment concentrations are necessarily an indicator of 'coastal processes'. If, for example, there is an offshore mud bank in shallow water, this may produce a surface signature of suspended sediment which is nothing to do with the coast. In the Baltic, the width of the measured coastal zone is greater in the south-east than it is in the north-west, a feature the authors explain in terms of the muddy nature of the sea bed in the south-eastern study site. I'm not sure that a muddy sea bed can be attributed to 'coastal processes'. So, a relevant question is where do the sediments come from? The coast or the sea bed? In my experience, in north-west Europe, the major source of fine sediments is the sea bed - they are not really associated with coastal processes at all. Things may be different in the Baltic. I would be interested in the author's views on that point.

My second concern is that the curve-fitting employed by the authors is rather blind and appears to ignore important features in the data. Curves are fitted to plots of suspended load against distance offshore and a threshold is identified which marks the edge of the coastal zone. But if we look at a plot like figure 11b (SE3) (for example) there is obviously a sharp drop in sediment load at a distance 5 km offshore which appears to me to mark the edge of something. But the curve fitting technique identifies the actual limit of the coastal zone as further offshore than this. An even clearer example is in figure 11b S4 where there is obviously a clear edge at nearly 10km offshore which is missed by the curve-fitting technique. So the curve-fitting is ignoring obvious features (fronts and edges) in the data. Author comment please.

minor comments:

abstract, line 8, 'geographical extent' may be a better term than 'physical extent' here.

section 2.1 line 2, we are told that the study areas are selected on 'run-off, depth, wind exposure and bottom substrates'. Some details of what these features are like at the different study sites would be useful here.

[Figure]

You seem a little unsure in your description of how you avoided times of cyanobacteria blooms. I know these are dominant in satellite images of the Baltic in summer and it's important to avoid them. I'm convinced when you get to the chlorophyll maps and tell us that these don't look like the TSM distributions. So, perhaps make that point right up front?

When you fit the polynomial curve, you choose the order of polynomial that gives the highest r-squared. Usually, r-squared will increase as the order of the polynomial increases, until eventually the curve goes through all the points. What was the upper limit of the polynomial you chose?

An important criterion in defining the width of a river plume entering the sea is the Rossby radius, which is the turning radius on a rotating earth. In a simple way, we can say that the coastal zone is likely to scale with the local Rossby radius. It might be worth mentioning this and telling us how the Rossby radius in the Baltic compares with the widths you measure.

Table 2 caption, please tell us what these are the 'min, max, mean and median values' of. Also, since the units of all are the same, perhaps the final column could go in the caption too?

When you are doing the polynomial and logarithmic curve fitting, it would be worth spelling out what exactly are the dependent and independent variables you are using. Presumably they are not the same for the two types of curve?

---

## Author Comment (AC1) · 27 Apr 2016

Thank you for your comments!

Yes, we agree that this is a rather descriptive article. The advantage of using satellite-derived TSM to define the extent of the coastal zone is that we can look at the whole Baltic Sea basin synoptically and visually compare the different coastal areas. But the work is not merely descriptive. We base our transect analysis also on theory developed by Kratzer and Tett (2009, see Box 1). Using bio-optical data from the NW Baltic Sea the authors showed how one case use bio-optical data to define the extent of the coastal zone (Figure 7).

[Figure]

In this current paper we used Kratzer and Tett (2009) as a basis of our analysis, but we use basin-wide remote sensing data instead to define the extent of coastal processes. We decided to talk here about the 'extent of coastal processes' rather than the extent of the coastal zone as we feel that SPM is more of an indicator of the processes influenced by the coast rather than an exact indicator of the coastal zone.

Our hypothesis is quite clearly defined in the introduction section and then three main objectives are stated at the end of the introduction:

Hypothesis: Total suspended matter derived from MERIS data can be used as an indicator of costal processes.

Objectives: Derive mean estimates of total suspended matter for the Baltic Sea and its sub-basins (please see Figure 12, page 27 and Table 2, page 17)

In section 3.1 the logic behind why we only used the early June images for the analysis is explained. Basically, we wanted to exclude the effect of cyanobacteria as we want to focus on the influence of coastal processes (page 6, Lines 17-22). Further, the TSM composites are compared to chlorophyll-a composites for the same dates illustrating different patterns and de-coupling of the two parameters (i.e. chl-a and TSM), indicating that the TSM values in the composite is presumably of inorganic origin (page 6, Lines 26-30) and followed by illustration (Figure 9).

Investigate if MERIS data can be used to evaluate the extent of coastal processes using bio-optical model derived from in situ measurements (that are essential for validation of satellite products) (please see page 24-26, Figure 10a, 11a & 11b)

Here, it needs to be explained that total suspended matter (TSM) which is also termed suspended particulate matter (SPM) can be divide into an organic and an inorganic fraction as described (page 5, Line 16-17). Kratzer and Tett (2009) used inorganic suspended matter to identify the break between coastal and open sea using a threshold of 0.05 ãĂŰgmãĂŮˆ(-3) which was reached at a distance of about 10-20 km from the

coast. As MERIS provides us with the total suspended matter as a standard product we had to estimate how 0.05 ãĂŰgmãĂŮˆ(-3) of inorganic matter translates into TSM. Using the bio-optical data described in Kratzer and Tett (2009; figures 4a, 7a&d) we could conclude that this threshold corresponds to 0.8 (+/- 0.3) ãĂŰgmãĂŮˆ(-3) TSM (now including both the organic and inorganic matter).

However, after a statistical analysis of a composite image from June 2011 for different coastal areas in the Baltic Sea we found that the overall threshold is closer to about 0.6 ãĂŰgmãĂŮˆ(-3) for total suspended matter (TSM). We found that this threshold is overall more representative for the Baltic Sea, but the values vary slightly dependent on the basin (see page 20-21, Figure 5 & 6). In the end we decided to use the respective local threshold between coastal and open sea waters to define the extent of the coastal in our trend analysis. The local threshold is then applied to the chosen transects for each area (see page 25-26, Figure 11a & 11b).

The extent of coastal processes is compared (synoptically) to the water body classification defined by SMHI (see section 3.2, page 7, Lines 5-31 this for how objective was described and page 24-25, Figure 11a & 1b achieved)

The reviewer was very right to point out that the errors in deriving TSM using the FUB algorithm is actually higher (about 27%) than when using the standard algorithm MEGS (about 10%, see Beltran-Abaunza et al. (2014). We have now corrected this in the revised paper. FUB is not the standard MERIS processor; but it has shown to be overall best for the retrieval of water products.

However, we feel that it is important to mention that the TSM concentration derived by MEGS (the standard processor) were associated with significant noise, whilst the FUB images showed much less noise. Additionally, FUB has a more consistent off-set over the reflectance spectrum (Beltran et al, 2014). FUB is also more accurate in the chlorophyll retrieval that gives us a degree of certainty that what we see in early June composite 2011 is suspended matter of inorganic origin, which is associated with river

run-off and coastal processes, which once again shows that the extent of coastal processes is way beyond earlier assumed scales (e.g. as commonly defined by the SMHI method). And as final remark, the choice of whether using FUB or MEGS for retrieval of TSM in a given area must take into account local conditions and ranges of in situ retrieved values. We have described the relative errors (RMS) and systematic (MNB) errors already in a previous study (Beltran-Abaunza et al., 2014), and even though the transect data do include these errors we still can extract the relative difference between coastal and open sea waters.

Additionally, we set approximate values (number in km) of how far the suspended matter reaches off-shore that is within similar ranges of values as the in situ based bio-optical model described. Here, satellite images allowed to extend the coastal transect further off-shore than the usual measurements from research vessel that generally, do not reach so far off-shore and are also rather sparsely distributed. The MERIS images indicated that the extent of coastal processes is substantially further than 1 nautical mile (as defined by the WFD). Coastal processes also extend somewhat further than the distance defined by in situ bio-optical model described in Kratzer and Tett (2009), especially if one includes Bråviken (in the west, NW2) in the analysis, where there is a much higher gradient of SPM than in the Himmerfjärden area. Our method based on satellite data also shows that coastal processes do extend further than the coastal waters defined by the water body classification developed by SMHI.

Our results demonstrate the advantage of using remote sensing data complementary to conventional monitoring methods, allowing us to rethink how to define the extent of coastal processes and potentially not only from national, economical, geographical but as well from an ecological perspective. This is extremely relevant for such a complex water body as the Baltic Sea with significant anthropogenic load and surrounded by a large number of countries.

Final reviewer's comment: "Finally, the authors have provided illustrations of the seabed sediment (Fig. 13) and bathymetry (Fig 14) in order to illustrate points in the

discussion. However, this information is not new and was not used in the methodology and result section."

We used already existing information (i.e. seabed sediment and bathymetry) to support our points in the discussion. This information was not produced by the authors and therefore was not new and on purpose was not included as part of methodology or results. We still considered it valuable data for our discussion.
* * *

---

## Author Comment (AC2) · 27 Apr 2016

Thank you David for your valuable comments and feedback! We found it to be relevant and on point.

To the first weakness of the paper we would like to respond as follows:

We agree that we have not really discussed the issue of mud banks, and we are now including this in the revision. Most of the mud banks are situated in the southern Baltic Sea and are rather close to the coast, so they can also be regarded as rather coastal features. During the ice age the glacial ice sheet went right across Scandinavia and reached also over most of the Baltic Sea, apart from the very south-eastern coastal

[Figure]

areas where we today find more sand banks. The rest of the Baltic Sea bottoms were swiped clear of muddy and sandy bottoms, so most of these features can still be found in the SW. But the origin of this matter is most likely also from coastal run-off and erosion. The suspended sediment is carried into the sea with river run-off can therefore be considered part of costal processes. Suspended matter also limits the light that penetrates the water, thus affecting primary production that in turn affects the ecology within coastal areas.

In the SE Baltic there seems to be a combination of both sediments brought in by river run-off which then is mixed up with sediment brought up from the sea bottom and carried away by wind and Coriolis force further offshore. If a muddy sea bed is situated close to the coast and thus contributing to the sediment loads through resuspension then it should also be natural to assume that they are part of 'coastal processes'. However, when looking at the SE part of the Baltic along the transect that we have extracted and evaluating the bathymetry (page 29, Figure 14) it is evident that there are no 'offshore mud banks' that may produce a surface signature. The sea bottom here is already rather deep close to the coast and gets even deeper when moving further off-shore into the open sea. There is presumably no way to distinguish from the satellite images whether suspended sediment was brought up from the bottom or if it is derived from run-off. That is why we refer to the bottom substrate map and bottom topography as a possible explanation why the extent of suspended matter is so wide in the SE Baltic. It is likely, though, that the sandy and muddy areas in the sediments also have been slowly built up over time by coastal run-off which is much stronger in the southern parts of the Baltic Sea than in the NW parts.

Second weakness: curve-fitting procedure

The initial bio-optical model described by Kratzer & Tett identifies the break between coastal and open sea at a point where SPM concentration reaches the threshold of about 0.6 ãÅŰgmãÅŮˆ(-3). Where the decay was best described by polynomial of 2nd order for SPM (first order polynomial for chl-a, and third order polynomial for CDOM).

This model was based on theory (diffusion is the driving force) and tested on in-situ optical measurements (see Box 1 and Figure 7 in Kratzer and Tett, 2009). So, we only considered 1st, 2nd and 3rd order polynomials but in general the 2nd order was the best polynomial description for SPM as found by Kratzer and Tett (2009). But we also found that in some cases a logarithmic described the trend better than any of the tested polynomials (especially in the south-eastern and southern Baltic). We concluded that this has to do with the stronger effects of wind-driven processes in the SE Baltic Sea, whereas in the NW Baltic diffusion may be assume to be the driving process for the distribution of particles. There are substantially more points extracted from a satellite image along a given transect than usually measured in situ. So, the degree of freedom is not really much affected by going from 2nd to a 3rd order polynomial. But in none of the cases the third order polynomial was chosen as it did not show a significantly improvement of the coefficient of determination.

The values from the MERIS transect has a rather high spatial resolution (300 m pixels) and highlights the features that would most likely have been missed by only using in situ transect data. The sudden sharp drops in values close to the shore can be indicators of processes that happen very close to the coast and are most likely closely linked to the water depth. When the water depth increases the values drop rapidly since there is no longer any contribution from the bottom and only a fraction of suspended sediments gets transported further away by wind or swell. It may well be that the depth profiles (see Figure 15) do not have the required spatial resolution to actually map the sea-floor correctly and include all information on coastal sand banks. A lot of this data counts as secrete military information and to date bottom maps with high resolution are not officially available.

Our rather generalized approach excluding obvious meso-scale features was required to be able to apply one model to different parts of the Baltic Sea so the results would be intercomparable and the overall coastal trends could be evaluated. So, when choosing transects for the trend analysis of the particle distribution we avoided large meso-scale

features on purpose, as the main goal was to evaluate the spatial extent of suspended particles rather than describing separate suspended features that may indicate certain oceanographic features such as large-scale eddies. In our choice of transects we thus assumed meso-scale features to be related to the internal circulation of the Baltic proper caused by the Coriolis force creating anti-clockwise currents in the Baltic proper basin, and thus not really directly indicating coastal features. But the images from the SW Baltic show that there is somewhat an overlap of the coastal sediment gradient with several of these features. The baroclinic Rossby radius varies between 1.5-10 km (Fennel et al., 1991; Alenius et al., 2003) and the costal upwelling is usually found at distances of 10-20 km offshore (Lehmann and Myberg, 2009). So, some of the features in the SW Baltic may also be caused by coastal upwelling moving suspended matter from the sea bottom up to the surface.

Yes, 'the geographical extent of coastal processes' would be more appropriate than only 'physical extent' as suggested by the referee.

We thought that chlorophyll maps should be brought up in the 'Results' section right next to the TSM composite from early June to highlight that chlorophyll patterns do not match those of the total suspended matter, therefore visually confirming that TSM concentration are of inorganic origin. Please see (page 6, Lines 17-25) where we additionally refer to Kahru & Elmgren (2014) who described the shift in the timing of cyanobacterial surface accumulations on average by 0.6 days per year, causing the peak of the summer bloom in recent years occurring 20 days earlier when compared to 1979.